# The Nutritional Profile and On-Pack Marketing of Toddler-Specific Food Products Launched in Australia between 1996 and 2020

**DOI:** 10.3390/nu14010163

**Published:** 2021-12-30

**Authors:** Jennifer R. McCann, Catherine G. Russell, Julie L. Woods

**Affiliations:** Institute for Physical Activity and Nutrition (IPAN), School of Exercise and Nutrition Sciences, Faculty of Health, Deakin University, Geelong, VIC 3220, Australia; georgie.russell@deakin.edu.au (C.G.R.); j.woods@deakin.edu.au (J.L.W.)

**Keywords:** ultra-processed, nutrition, public health, toddler, food environment

## Abstract

With the food system evolving, it is not clear how the nutrition and on-pack claims of toddler foods have been impacted. Data on the trends in Australia are lacking, so we sought to determine the changes in the nutrition and on-pack claims of toddler-specific packaged foods over time. A retrospective cross-sectional analysis was conducted using the Mintel Global New Products Database. The number of toddler-specific foods increased from 1996 to 2020. Over time, a lower proportion of meals and snacks were classified as “ultra-processed”, but a higher proportion of snacks were classified as “discretionary”. Meals launched after 2014 had higher median values for energy, saturated fat, and sugar than those in earlier years. Toddler snacks launched after 2014 had lower median values for sodium, and higher median values for fat, saturated fat, and sugar than those in earlier years. The mean number of total claims per package increased over time for snacks, with an increase in unregulated claims for both meals and snacks. Public health action is needed to ensure that the retail food environment for young children is health-promoting, including stringent and clear regulations for on-pack claims, and compositional guidelines and guidance on how to reduce the number of ultra-processed foods for toddlers.

## 1. Introduction

In Australia, poor diet quality and childhood overweight and obesity are major public health issues [1,2,3]. The causes of these problems are multifactorial; however, the food environment is increasingly recognised as an important contributor [4,5,6]. Because of a major shift in the global food system in Western countries such as Australia, ultra-processed (UP) and discretionary foods contribute close to 50% of the total dietary intake in young children [3,7,8,9,10]. There is strong evidence on the harms of UP food consumption on the paediatric population [8,11,12], which include impeding continued breastfeeding [8], overweight and obesity [13], and altering the taste palate [14,15].

A key aspect of the food system is on-pack marketing, which influences consumer purchases of UP and discretionary foods [16,17,18,19,20,21]. Consumers are often confused by claims and are potentially being misled by on-pack claims into thinking foods are healthier than they really are [22,23,24]. While it is known that the food system has evolved over time, it is unclear how the nutrition profile and on-pack marketing of toddler foods has changed as a part of this. With real-world data on trends over time in the retail toddler food space in Australia lacking, the aim of this study was to determine the changes over time in the nutritional profile and on-pack claims in toddler-specific packaged foods launched in the Australian retail market.

## 2. Materials and Methods

The Mintel Global New Products Database (Mintel) was searched for the following predefined categories: baby biscuits and rusks; baby cereals; baby fruit products, desserts and yogurts; baby juices and drinks; baby savoury meals and dishes; baby snacks; and other baby food, in the Australian retail market from June 1996 (inception of Mintel) to December 2020 (no foods are categorised as “toddler” in the Mintel database). All product images, nutrients from the nutrition information panel, ingredient lists, and on-pack claims were exported into an MS Excel file. After exporting, to determine their inclusion as toddler food (1–3 years), the product images, information, and descriptions were manually checked for the age range they were being marketed to.

All on-pack claims were counted and subclassified as “regulated” and “unregulated”, as per Schedule 4 of the Food Standards Australia New Zealand (FSANZ) Food Standards Code [25]. Regulated claims included nutrition-content claims (e.g., “no added sugar”) and health claims (e.g., “calcium for strong bones”), while unregulated claims included health-related ingredients (e.g., “no added preservatives”), natural/organic claims (e.g., “all natural” or “certified organic”), environmental claims (recycled logo), and others (e.g., “perfect for small hands”).

Products were classified in three ways: as “meals” or “snacks”, based on the categories identified in a 2019 World Health Organization report [26]; as “core” or “discretionary”, based on the Australian Dietary Guidelines [27] and the Australian Bureau of Statistics discretionary food list [28]; and by the level of processing as per the NOVA classification (ultra-processed (UP), processed (P), and minimally processed (MP)) [29].

### Data Analysis

All analyses were conducted using SPSS V26. Data were split into time quartiles (T1 (1996–2002), T2 (2003–2008), T3 (2009–2014), and T4 (2015–2020)) to explore the changes over time. Meals and snacks were analysed separately because of the different nutritional profiles and classifications (meals are classified as “core” products according to the ADG, while many snacks are not). For each time period, descriptive analyses were performed to determine the number and proportions of the products launched, the food type classifications, the median nutrient values, and the means and ranges of different types of claims. Kruskal–Wallis tests and ANOVA tested for differences in the median nutrient values and mean claim types across each time quartile, while Chi-square tested for the ADG and NOVA classification significance across time periods. The significance levels were set at the acceptable minimum significance level of <0.05.

## 3. Results

### 3.1. Total Products

A total of 213 products (39 meals and 174 snacks) were identified. Of these, ten had no photos or descriptive data (nine in T1 (eight snacks and one meal), and one snack in T2), and so these were excluded from all analyses, except the total number of products. In T1, there were two meals and eight snacks. There was one meal in T1 with photos and descriptive data, and this was a core and UP product. More detail for time-periods 2–4 can be seen in Table 1. The majority (82%) of products were snack foods. There were equal proportions of core (50%) and discretionary (50%) foods, and 76% of all products were classified as UP.

### 3.2. Results over Time

#### NOVA and Australian Dietary Guidelines Classifications

More products were launched in each time period compared to the previous, as seen in Table 1. Snacks showed some fluctuation in classification within the ADG and NOVA over time: In T2, 41% of snacks were classified as “core”, and 88% were classified as “UP”; in T3, 52% were classified as “core”, and 88% were classified as “UP”; and in T4, 30% were classified as “core”, and 78% were classified as “UP”. The changes in the classifications of snacks by the ADG and NOVA across time periods were statistically significant (both *p* < 0.001). The proportion of meals classified according to the ADG did not change over time, with all meals being classified as “core”; however, there were changes in the NOVA classification, with a reduction in the proportion of products classified as “UP” between T2 (86%) and T4 (45%) (*p* < 0.05).

### 3.3. Nutritional Characteristics

The median nutrient values per 100 g for both meals and snacks during time-periods 2–4 are shown in Table 2. For meals, there were no significant differences in the median nutrient values in each time period. For snacks, there were significant differences in the median energy, total fat, and saturated fat values across the time periods (all *p* < 0.05), although the differences are very small and are unlikely to be nutritionally important.

### 3.4. On-Pack Claims

The total numbers, mean numbers, and ranges of each type of unregulated and regulated claims across the time periods are detailed in Table 3. For meals, the only difference in the mean numbers of claims were related to unregulated environmental claims, with all other regulated and unregulated claims not demonstrating any differences across the time periods. The mean number of unregulated health-related ingredient claims, natural/organic claims, and other claims were significantly different across the time periods for snacks (all *p* < 0.05).

## 4. Discussion

Our results show that there were relatively more toddler-specific packaged foods (meals and snacks) launched onto the Australian retail market at each of the four time periods investigated. Notably, substantially more toddler-specific snack foods were launched during T3 (2009–2014) and T4 (2015–2020) than in T1 (1996–2002) and T2 (2003–2008). In relation to the healthiness of foods launched during the time periods, the findings were mixed, with a lower percentage of meals classified as “UP” in T4 than in earlier time periods, and a lower percentage of snacks classified as “UP” and “core” during T3 and T4 than in T2. While the proportion of UP toddler-specific foods on the Australian retail market decreased over time, this was paralleled by an increase in toddler snacks classified as “discretionary”. The mean number of total claims remained relatively stable over time on meals but was higher in T3 and T4 compared to T2 on snacks, which is due to an increase in unregulated claims.

The finding of a greater number of new products launched in T4 compared to other time periods concurs with Australian and global retail data, which shows an expansion of the toddler-specific food and milk markets over time [30,31]. The majority of new products (74%) launched over the 25-year timeframe of this study were UP. This aligns with analyses of the current toddler retail market, which reflect the cumulative effects of product launches and cessations over time, and that have shown that most toddler-specific foods in the retail market are toddler-specific packaged snacks and are UP [32,33]. We found a large decrease in the percentage of meals classified as “UP” from T2 (86%) through to T4 (45%); however, all meals across all time periods were classified as “core” foods. In addition, despite the lower percentage of UP snacks launched in T4, which may represent an acknowledgement of the harms associated with these foods by manufacturers, there was a higher percentage of discretionary snacks launched at the same time. The food environment, therefore, still favours foods not recommended for toddlers.

That the toddler retail market is largely UP and discretionary is concerning, as the harmful effects of a diet high in UP and discretionary foods on the health of the paediatric population are well documented [8,13,14,34,35]. Despite the percentage of UP toddler-specific foods decreasing from T2 (84%) to T3 (83%), overall, 74% of all products at T4 were still UP. Paralleling this decrease in the percentage of UP products, the percentage of discretionary snack foods increased at each time point from T2 to T4. At a young age, the normalisation of UP and discretionary foods, which are often salty, fatty, or sweet, and homogenous in texture [11,33,36], can have detrimental impacts on the formation of eating behaviours, and can have ongoing effects on diet and health. [36,37,38] Instead, young children should be exposed to a variety of tastes, flavours, and textures in the critical early years to develop healthy eating habits [39,40].

Nutrient composition across all products appears to have remained similar over time, despite changes in the NOVA classification. This may be partly explained by the fact that the predominant type of snack food was fruit-based cereal bars, and some reformulation over time may have occurred. The small changes in nutrient compositions seen may align with product reformulation within this retail market [41,42,43,44], as snacks were seen to have lower median sodium, sugar, and carbohydrate levels at T3 as compared to T2, with sugar increasing at T4. Global research over various time periods reports toddler foods to be high in sugar and/or sodium [45,46,47,48,49]. In addition, an Australian study reported that from 2009–2011, 67% of child-oriented food products underwent reformulation, with 15% having simultaneous positive (e.g., reduction in sodium) and negative (e.g., increase in sugar) reformulations, and overall little improvement in healthiness. [50] Within the food industry, the current state of practise is very nutrient-centric [51], with most reformulation efforts focusing on reducing sugar or sodium levels. However, there are limits to which nutrients can be reduced within a UP food before further additives are required to retain the hyper-palatability of the food. This nutrient-centric view of reformulation appears to be promoting the consumption of UP food through on-pack claims and messaging [52].

Claims on products remained quite similar across all products and all time periods, apart from the unregulated claims relating to no additives, organic claims, and “other” claims (claims related to being “good for small tummies”, “easy to hold”, or story-based messaging about the company). Despite the seemingly small changes in the number of claims on these products, the mean number of claims (and the maximum number of claims) per product was high for both meals (mean = 5, max = 12) and snacks (mean = 8, max = 14), and is comparable to research from the United States on infant and toddler foods [53]. Evidence shows that baby and toddler packaged foods often include numerous on-pack claims, both regulated and unregulated (depending on the country) [32,53,54]. Claims on food packaging have been shown to influence consumer purchasing [16,22,55,56], and they can create an increase in health perceptions for many products [16,57]. This can then mislead consumers, as unhealthy foods can appear healthier by their presence, for instance, by only making claims about positive nutrient characteristics (e.g., “low sugar” or “low sodium”), while being high in nutrients of concern (e.g., the product may be low-sugar but have a high sodium content). The majority of these foods are either discretionary choices or UP, and are, therefore, not recommended for toddlers, and yet companies are permitted to display these claims. Indeed, Australian food regulation permits the use of nutrient content claims on all foods, regardless of the overall nutrition profile or the level of processing.

## 5. Limitations

This study provides some insights into the practices of toddler food manufacturers over time but does not enable a thorough and detailed examination of the whole toddler food environment at all points along the timeframe covered. It is possible that some foods may have been missed within the Mintel database because of the manual nature of identifying toddler foods. The classification of foods by ADGs and NOVA can be difficult occasionally, but a 10% sample of products was double-checked for accuracy to help limit misclassifications, of which there were none.

## 6. Conclusions

Our results demonstrate that, despite a decrease over time in the percentage of foods classified as UP, toddler-specific foods being launched into the Australian retail market are highly processed (and have been since 1996, at least), and toddler snack foods are mostly discretionary choices, and are not recommended by the ADG. Despite these results, these products are being promoted to parents of young children, often with on-pack claims indicating that they are healthy. To promote the launch of healthier toddler foods onto the Australian retail market, public health action is needed. This includes stringent and clear labelling regulations for on-pack claims, clear guidelines to reduce the number of UP foods being marketed for toddlers, and compositional guidelines for foods made for toddlers.

## Figures and Tables

**Table 1 nutrients-14-00163-t001:** Number of products launched in Time-periods 2-4, and classifications within ADG and NOVA.

	Meals			Snacks			Total Products		
	*n*	ADG ^1^ Core *n* (%)	NOVA UP ^2^* *n* (%)	*n*	ADG Core ** *n* (%)	NOVA UP ** *n* (%)	*n*	ADG Core ** *n* (%)	NOVA UP ** *n* (%)
Time-period 2 (2003–2008)	7	7(100)	6(86)	17 ^3^	7(41)	15(88)	25	14(56)	21(84)
Time-period 3 (2009–2014)	19	19(100)	13(68)	58	30(52)	51(88)	77	49(64)	64(83)
Time-period 4 (2015–2020)	11	11(100)	5(45)	90	27(30)	70(78)	101	38(38)	75(74)

* *p* < 0.05, ** *p* < 0.001, χ^2^. T1 excluded because of no available on-pack data. ^1^ ADG: Australian Dietary Guidelines; ^2^ UP: ultra-processed. ^3^ Total numbers have been adjusted to exclude the products without photos or descriptive data.

**Table 2 nutrients-14-00163-t002:** Median nutrient values/100 g and (interquartile range) of meals and snacks launched in time-periods 2, 3, and 4.

	Meals T2 *n* = 7	Meals T3 *n* = 19	Meals T4 *n* = 11	Snacks T2 *n* = 17 ^1^	Snacks T3 *n* = 58	Snacks T4 *n* = 90
Energy (kJ)	276 (247)	304 (150)	395 (183)	1600 (270)	1481.7 (1275)	1657 * (382)
Protein (g)	3.9 (6)	3.5 (2)	3.9 (3)	7.1 (6)	5.2 (5)	5.5 (5)
Total fat (g)	2.9 (9)	2.0 (1)	2.1 (2)	6.3 (9)	3.7 (8)	6.5 * (14)
Saturated fat (g)	0.5 (1)	0.6 (1)	0.8 (2)	2.3 (3)	1.2 (3)	2.5 * (3)
Carbohydrate (g)	9.4 (5)	9.8 (4)	9.2 (4)	65.1 (22)	65.9 (58)	65.9 (18)
Sugar (g)	2.3 (3)	2.5 (1)	2.5 (1)	19.3 (31)	14.8 (30)	23 (34)
Sodium (mg)	60 (55)	30 (75)	55 (70)	95 (234)	73 (194)	32 (192)

* *p* ≤ 0.05 for across all time periods. T1 excluded because of no available on-pack data. T2: Time-period 2 (2003–2008); T3: Time-period 3 (2009–2014); T4: Time-period 4 (2014–2020). ^1^ Total numbers have been adjusted to exclude the products without photos or descriptive data.

**Table 3 nutrients-14-00163-t003:** Mean numbers (ranges) of claims per product by type for meals and snacks launched in time-periods 2, 3, and 4.

	Meals T2	Meals T3	Meals T4	Snacks T2 ^1^	Snacks T3	Snacks T4
	*n* = 7	*n* = 19	*n* = 11	*n* = 17	*n* = 58	*n* = 90
Unregulated health-related ingredient claims (*n* = 774)	3 (1–5)	3 (0–6)	3 (1–6)	3 (0–7)	4 (0–8)	4 (0–10) *
Unregulated natural/organic claims (*n* = 102)	0 (0–2)	1 (0–1)	1 (0–1)	0 (0–2)	0 (0–2)	1 (0–3) *
Unregulated environmental claims (*n* = 101)	0 (0)	0 (0–1)	0 (0–2) **	0 (0–1)	1 (0–2)	1 (0–3)
Unregulated “other” claims (*n* = 96)	0 (0)	0 (0–1)	0 (0–5)	0 (0–1)	0 (0–1)	1 (0–7) *
Regulated nutrition-content claims (*n* = 216)	2 (0–5)	1 (1–4)	1 (0–4)	1 (0–4)	1 (0–4)	1 (0–4)
Regulated health claims (*n* = 33)	1 (0–5)	0 (0)	0 (0)	0 (0–1)	0 (0–6)	0 (0–3)
Total unregulated claims (*n* = 1073)	3 (1–6)	3 (0–7)	3 (5–8) *	4 (0–9)	5 (0–10)	7 (2–13) **
Total regulated claims (*n* = 249)	3 (0–10)	1 (0–4)	1 (0–4)	1 (0–4)	1 (0–10)	1 (0–4)
Total claims (*n* = 1322)	6 (3–11)	5 (0–9)	5 (5–12) *	5 (0–10)	6 (0–18)	8 (2–14) **

* *p* < 0.05, ** *p* < 0.001 between time periods. T1 excluded because of no available on-pack data. T2: Time-period 2 (2003–2008); T3: Time-period 3 (2009–2014); T4: Time-period 4 (2014–2020). ^1^ Total numbers have been adjusted to exclude the products without photos or descriptive data.

## Data Availability

Mintel database is not publicly available; however, select data for research purposes can be made available upon request to the corresponding author.

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
