# Peer review of "The Nutritional Profile and On-Pack Marketing of Toddler-Specific Food Products Launched in Australia between 1996 and 2020"

_nutrients, 2021, doi:10.3390/nu14010163_

Round 1

Reviewer 1 Report

The methodology and its limitations are clear. However, I do not clearly identify how you classified the products, since you refer (line 57): Products were classified as a snack or meal… snacks were subclassified according to their main ingredient and aligned with methods by de WHO (I do not identify the sub classification). Products were also classified as core or discretionary according to the ADG… what was the procedure to classify them?

The different names are a bit confusing, in case each one could be reduced or made very clear.

Please consider another limitation: it is not known if the values ​​of the nutrient content in 100 g are adequate per serving for children.

Line 144: there is no basis to assume, since although UP snacks decreased, there was a high percentage of discretionary snacks in the same period.

Reviewer 2 Report

  • How did you set the significance level? 
  • Unit of sodium is mg instead of g (Table 2)?
